# A Modified Flotation Density Gradient Centrifugation Technique Improves the Semen Quality of Stallions with a High DNA Fragmentation Index

**DOI:** 10.3390/ani11071973

**Published:** 2021-07-01

**Authors:** Muhammad Umair, Heiko Henning, Tom A. E. Stout, Anthony Claes

**Affiliations:** 1Department of Clinical Sciences, Faculty of Veterinary Medicine, Utrecht University, Yalelaan 112, 3584 CM Utrecht, The Netherlands; Heiko.Henning@fli.de (H.H.); T.A.E.Stout@uu.nl (T.A.E.S.); A.Claes@uu.nl (A.C.); 2Friedrich-Loeffler-Institut, Federal Research Institute for Animal Health, Höltystraße 10, D-31535 Neustadt, Germany

**Keywords:** stallion spermatozoa, sperm selection, intact DNA, density gradient centrifugation, Opti-prep^TM^

## Abstract

**Simple Summary:**

Sperm DNA fragmentation has a negative impact on reproductive success, because it compromises fertilization and embryo development. Since spermatozoa lack the machinery to repair DNA damage, the likelihood of establishing a healthy pregnancy can be improved by processing ejaculates with a high sperm DNA fragmentation index. The current study examined a modified flotation density gradient centrifugation technique in which semen was diluted with a colloid solution (Opti-prep^TM^) to increase its density, later layered between colloid layers of lower and higher density. This modified flotation technique was consistent in selecting viable, progressively motile, morphologically normal spermatozoa with intact acrosome and DNA. In reproductively normal stallions, sperm recovery was 57% and spermatozoa with damaged DNA decreased significantly (from 12% to 4%), with increase in the percentage of viable, progressively motile, morphologically normal spermatozoa with intact acrosome. In stallions with a high sperm DNA fragmentation index, this modified flotation technique markedly decreased the DNA fragmentation index (from 31% to 5%) and significantly improved the other semen quality parameters, although sperm recovery was low (approximately 20%). In conclusion, the stallion sperm DNA fragmentation index and other sperm quality parameters can be markedly improved using a modified flotation density gradient centrifugation technique employing a 40% Opti-prep^TM^ cushion and a 20% top layer.

**Abstract:**

Sperm DNA fragmentation compromises fertilization and early embryo development. Since spermatozoa lack the machinery to repair DNA damage, to improve the likelihood of establishing a healthy pregnancy, it is preferable to process ejaculates of stallions with a high sperm DNA fragmentation index (DFI) before artificial insemination or intracytoplasmic sperm injection. The aim of this study was to examine a modified flotation density gradient centrifugation (DGC) technique in which semen was diluted with a colloid solution (Opti-prep^TM^) to increase its density prior to layering between colloid layers of lower and higher density. The optimal Opti-prep^TM^ solution (20–60%) for use as the bottom/cushion layer was first determined, followed by a comparison between a modified sedimentation DGC and the modified flotation DGC technique, using different Opti-prep^TM^ solutions (20%, 25% and 30%) as the top layer. Finally, the most efficient DGC technique was selected to process ejaculates from Friesian stallions (*n* = 3) with high sperm DFI (>20%). The optimal Opti-prep^TM^ solution for the cushion layer was 40%. The modified sedimentation technique resulted in two different sperm populations, whereas the modified flotation technique yielded three populations. Among the variants tested, the modified flotation DGC using 20% Opti-prep^TM^ as the top layer yielded the best results; the average sperm recovery was 57%; the DFI decreased significantly (from 12% to 4%) and the other sperm quality parameters, including progressive and total motility, percentages of spermatozoa with normal morphology and viable spermatozoa with an intact acrosome, all increased (*p* < 0.05). In Friesian stallions with high sperm DFI, the modified flotation DGC markedly decreased the DFI (from 31% to 5%) and significantly improved the other semen quality parameters, although sperm recovery was low (approximately 20%). In conclusion, stallion sperm DFI and other sperm quality parameters can be markedly improved using a modified flotation DGC technique employing a 40% Opti-prep^TM^ cushion and a 20% top layer.

## 1. Introduction

Sperm DNA fragmentation has a negative impact on reproductive success, because it compromises fertilization and embryo development, leading to a lower likelihood of pregnancy and a higher incidence of pregnancy loss [1]. The integrity of sperm DNA can be assessed using the sperm chromatin structure assay (SCSA), in which the percentage of sperm with increased levels of single-stranded DNA is represented by the DNA fragmentation index (DFI) [2]. In stallions, there is a negative association between the DFI and seasonal pregnancy rate, and the DFI is significantly higher in subfertile than fertile stallions [3,4]. Although sperm motility and morphology are often reduced in the ejaculates of stallions with high sperm DFI, the general semen quality parameters sometimes fail to reflect a high sperm DFI [5]. Furthermore, subfertile stallions with high sperm DFI and/or poor semen quality parameters are not universally excluded from commercial breeding programs, because stallions are usually selected predominantly on the basis of their athletic performance compared to other domestic species (bulls and pigs) and pedigree rather than fertility potential. Since sperm cells lack the machinery to repair DNA damage [6], and sperm DNA fragmentation is thought to be a non-compensable defect [7], in instances of high sperm DFI, it is considered important to reduce the number of spermatozoa with a relatively high degree of fragmented DNA by applying a sperm selection technique prior to assisted reproductive technologies (ART) such as artificial insemination (AI), conventional in vitro fertilization (IVF) or intracytoplasmic sperm injection (ICSI).

Density gradient centrifugation (DGC) is a sperm selection technique that separates spermatozoa into subpopulations based on their density. This is accompanied by a relative enrichment of viable and/or motile sperm with more uniform morphology in one of the subpopulations [8]. For equine assisted reproduction, single- or double-layer density gradient systems predominate among commercially available products (for a review, see reference [9]). In this context, Opti-prep^TM^ (Iodixanol 60%), originally developed as an X-ray contrast medium, is one of the colloid solutions that has previously been used to separate spermatozoa from men [10,11], bulls [8] and stallions [12]. Opti-prep^TM^ is nonionic, iso-osmotic and extensively tested for toxicity towards biological materials and can be used to generate isotonic solutions with densities up to 1.32 g/mL [13]. This property of Opti-prep^TM^ makes it suitable as a bottom or “cushion” layer to prevent the over-compaction of the sperm pellet during centrifugation. Less-concentrated solutions of Opti-prep^TM^ can serve as the top layer in double-layer DGC, which separates spermatozoa into populations with different qualities [8,12]. Thus, discontinuous double-layer DGC using Opti-prep^TM^ [12] to make both a cushion layer and a selective top layer has been proposed to be superior to ordinary centrifugation or density gradient centrifugation systems (single or double-layer) using Percoll [14], EquiPure [15] or Androcoll [16], because, in the latter, the selected sperm are collected as a pellet at the bottom of the tube. Pelleting onto the bottom of the tube is thought to be deleterious, because the sperm are tightly packed together with the debris and reactive oxygen species released from dying cells [9,10,17].

Previously, a double-layer DGC method involving layering diluted semen onto a top layer of low-density Opti-prep^TM^, which serves to select out less dense sperm, and underlain by a bottom layer of high-density Opti-prep^TM^, which serves as a cushion [12], has been used to process stallion semen. However, in this classical approach, sperm recovery is low (33%) even for ejaculates from stallions with semen parameters within the normal range. An alternative method of DGC (modified flotation DGC) has been described for bull semen, where the semen is first diluted (1:1) with a gradient solution (Opti-prep^TM^) to increase its density and then layered beneath two layers of Opti-prep^TM^ with decreasing density [8]. Whether and to what extent this different DGC setup may improve sperm recovery and the selection of sperm with intact DNA is not clear.

The aim of the current study was to systematically develop a DGC technique based on Opti-prep^TM^ that consistently separates out sperm cells with damaged DNA, both from stallions with normal semen quality parameters and stallions with high DFI, but with a minimum of other adverse effects on the sperm quality and recovery. This was performed in stages. First, the optimal Opti-prep^TM^ concentration to serve as a bottom/cushion layer was determined; this was followed by a comparison between a modified sedimentation and a modified flotation DGC using solutions of different densities. Finally, the modified DGC technique was used to process semen samples from a small number of Friesian stallions with very high sperm DFI (>20%) with the aim of enriching the selected population for progressively motile, morphologically normal spermatozoa with intact DNA.

## 2. Materials and Methods

### 2.1. Preparation of Different Density Gradient Solutions

Different density gradient solutions (20%, 25%, 30%, 40% and 50% Opti-prep^TM^ with calculated densities 1.110 g/mL, 1.137 g/mL, 1.163 g/mL, 1.215 g/mL and 1.268 g/mL, respectively) were prepared by diluting a 60% Opti-prep^TM^ stock solution (60% Iodixanol; Sigma-Aldrich Chemicals, Zwijndrecht, The Netherlands) with HBSS (Hanks Buffered Salt Solution; 5.33-mM KCl, 0.441-mM KH_2_PO_4_, 4.17-mM NaHCO_3_, 137.92-mM NaCl, 0.338-mM NaH_2_PO_4_ and 5.56-mM glucose, pH 7.4; Thermo Fisher Scientific, Ermelo, The Netherlands).

### 2.2. Semen Collection and Initial Processing

Semen was collected from stallions using a Hannover model artificial vagina. The volume of gel-free semen was measured, and the concentration of spermatozoa was determined using a NucleoCounter SP-100 (Chemometec, Allerod, Denmark). Subsequently, the semen was prediluted (1:1) with prewarmed (37 °C) INRA-96 extender (INRA; IMV Technologies, L’Aigle, France) and transported to the laboratory in a Styrofoam box.

### 2.3. Determining the Optimal Opti-prep^TM^ Solution to Use as the Bottom/Cushion Layer

Semen was collected from 6 stallions of different breeds (5 ± 1 years of age) and processed as described above. Semen diluted 1:1 in INRA-96 (2 mL) was then layered on top of a 20%, 30%, 40%, 50% or 60% Opti-prep^TM^ solution (2 mL) in a 15-mL centrifugation tube with a conical bottom (Greiner Bio-one, GmbH, Frickenhausen, Germany). The tubes were then centrifuged at 1000× *g* for 20 min at room temperature [12]. The proportion of sperm that passed through the cushion layer (sperm loss) was calculated by measuring the volume and determining the sperm concentration of the pellet that developed beneath the Opti-prep^TM^ solution after centrifugation, using a NucleoCounter SP-100 according to the manufacturer’s instructions and with the following formula:Sperm loss % = (total sperm in pellet ÷ total sperm loaded) ∗ 100

### 2.4. Modified Sedimenataion and Modified Flotation DGC Techniques

#### 2.4.1. Semen Collection and Processing

Following the collection of an ejaculate from 7 stallions of different breeds (6.2 ± 4.2 years of age) and the initial processing, as described above, the diluted semen was divided into two portions. One portion was further diluted (1:1) with INRA-96 (DS = diluted semen) and used for the conventional double-layer density gradient technique (referred to as the modified sedimentation DGC technique), i.e., diluted semen was layered on top of two solutions of different densities (Figure 1a). A sub-portion of the diluted semen was kept at room temperature to serve as a diluted, non-centrifuged control (NC) sample. The other prediluted aliquot was further diluted (1:1) with Opti-prep^TM^ 60% solution (DS-Opti-prep^TM^) to increase the density of the semen and used for the modified flotation DGC technique (Figure 1b). Semen samples recovered before (raw and diluted semen) and after centrifugation were snap-frozen for a subsequent DNA integrity analysis by the sperm chromatin structure assay (SCSA) and alkaline comet assay.

#### 2.4.2. Density Gradient Centrifugation (Modified Sedimentation vs. Modified Flotation DGC)

Three density gradient tubes (15 mL) were prepared for each of the modified sedimentation (Figure 1a) and the modified flotation DGC techniques (Figure 1b) by pipetting 2-mL layers of density gradient solution or diluted semen (DS vs. DS-Opti-prep^TM^). The order of the solutions from bottom to top was:(1)Modified sedimentation DGC (2 mL/2 mL/2 mL)
(a)Opti-prep^TM^ 40%/Opti-prep^TM^ 20%/DS(b)Opti-prep^TM^ 40%/Opti-prep^TM^ 25%/DS(c)Opti-prep^TM^ 40%/Opti-prep^TM^ 30%/DS
(2)Modified flotation DGC (2 mL/2 mL/2 mL)
(a)Opti-prep^TM^ 40%/DS-Opti-prep^TM^/Opti-prep^TM^ 20%(b)Opti-prep^TM^ 40%/DS-Opti-prep^TM^/Opti-prep^TM^ 25%(c)Opti-prep^TM^ 40%/DS-Opti-prep^TM^/Opti-prep^TM^ 30%

In comparison to the report on bull semen [8], the 1:1:1 mixture of stallion semen, INRA96 and Opti-prep^TM^ 60% (DS-Opti-prep^TM^) was less dense than pure 40% Opti-prep^TM^ and could not be placed on the bottom of the tube. Therefore, the DS-Opti-prep^TM^ layer was used as the intermediate layer. All tubes were centrifuged at 1000× *g* for 20 min at room temperature. After centrifugation (modified sedimentation and modified flotation DGC), semen bands were recovered separately from the interfaces (from top to bottom) with a 1-mL pipette, and the sperm concentration was adjusted using INRA-96 or HBSS according to the semen parameters to be investigated and kept at room temperature until further analysis.

#### 2.4.3. Semen Analysis

Sperm concentration, motility, viability, acrosome integrity, morphology and DNA integrity were determined prior to centrifugation and in all sperm bands after the modified sedimentation or modified flotation DGC techniques.

##### Sperm Recovery (SR %)

Sperm recovery was calculated by determining the sperm number in the initial sperm solution and in each band after modified sedimentation or modified flotation DGC by using the volume and concentration measured using a NucleoCounter SP-100 (Chemometec, Allerod, Denmark).

##### Sperm Motility

The following sperm motility and kinematic parameters were evaluated using a Computer-Assisted Sperm Analysis (CASA) system (Sperm Vision^®^; Minitüb, Tiefenbach, Germany): total motility (TM %), progressive motility (PM %), average path velocity (VAP μm/s), curved line velocity (VCL μm/s), straight line velocity (VSL μm/s), amplitude of lateral head displacement (ALH μm), straightness (STR % = VSL/VAP ∗ 100), linearity (LIN % = VSL/VCL ∗ 100), beat cross-frequency (BCF Hz) and wobble (WOB% = VAP/VCL ∗ 100). In short, semen samples (before and after centrifugation) were diluted to a concentration of 25–30 million/mL with INRA-96 and incubated at 37 °C for 5 min in a metal heating block; three microliters were then loaded into a 20-µm-deep Leja chamber slide (Leja Products BV, Nieuw-Vennep, The Netherlands). Sperm motility was examined in 12 sequential fields in the central part of the chamber, selected by an automated stage (30 frames/field at 60 Hz) using an Olympus BX41 microscope with a camera (resolution: 648 × 484 pixels, PulnixTM-6760CL; JAI A/S, Glostrup, Denmark) and an adaptor (U-PMTVC, Olympus, Hamburg, Germany) at 200× magnification and recorded using SpermVision^®^ software (Version 3.5.6.; Minitüb, Germany). Settings for the classification of motile and progressively motile spermatozoa were as described by Brogan et al. [18].

##### Viability and Acrosome Integrity (VAI %)

Sperm viability was evaluated using a combination of fluorescent stains, namely Hoechst 33342 (50 µg/mL; Sigma-Aldrich Chemicals) and propidium iodide (PI, 1 mg/mL; Thermo Fisher Scientific). Acrosome intactness was assessed using peanut agglutinin conjugated to Alexa Flour™ 488 (PNA-AF488: 1 mg/mL; Thermo Fisher Scientific). Briefly, 484 µL of HBSS and 10 µL of sperm suspension (5 × 105 spermatozoa) was pipetted into a flow cytometry tube to which was added 2 µL of each fluorescent stain, i.e., Hoechst 33342 (H33342), PI and PNA-AF488. The cell suspension was gently vortexed, incubated for 15 min at room temperature and examined (10,000 cells per sample) using a FACS Canto II flow cytometer equipped with 405 nm (30 mW), 488 nm (20 mW) and 633 nm (17 mW) lasers. Blue (H33342), green (PNA-AF488) and red (PI) fluorescence were collected by 450/50 BP, 530/30 BP and 585/42 BP filters, respectively. After defining the single-cell population on the basis of forward and side scatter, the sperm population was further gated to obtain the percentages of viable (H33342-positive and PI-negative) and acrosome-intact (PNA-AF488-negative) spermatozoa in FCS Express software (version 3; De Novo Software, Los Angeles, CA, USA). Spectral overlap was compensated after acquisition of the data.

##### Normal Morphology (NM %)

Semen samples prior to (raw) and after centrifugation (all bands) were fixed in buffered formol saline [19]. Differential interference contrast (DIC) microscopy was used to evaluate the morphology of the sperm cells. A total of 200 sperm cells per sample were evaluated for sperm defects. Each sperm cell was examined at 1000× magnification (oil immersion) and scored individually for all defects of acrosome (knobbed); head (detached, crater, vacuole and pyriform); neck (proximal droplet); mid-piece (distal droplet, bent, mitochondrial aplasia and double) and tail (coiled/bent and double) to classify it as morphologically normal or abnormal [20,21].

##### Sperm DNA Integrity

Sperm Chromatin Structure Assay (SCSA)

Semen samples collected prior to (raw) and after centrifugation (all bands) were snap-frozen and stored in liquid nitrogen. Sperm DNA integrity was evaluated using the sperm chromatin structure assay (SCSA) as described by Evenson and Jost [22]. Acridine orange stain was used after acid denaturation to identify the proportion of spermatozoa possessing unstable chromatin. Fluorescence signals (green and red) were collected using the 530/30 BP and the 670-nm-long pass filter, respectively, on the FACS Canto II flow cytometer. FCS express software was used to evaluate the recorded files. Ten thousand spermatozoa were recorded for each sample. Analysis quality assurance was attained by running a semen sample with the known DNA fragmentation index (DFI) before starting a flow cytometry session and after every twenty samples in a session. Total DFI% was determined by including cells with moderate and high DFI [23].

Alkaline Comet Assay

The comet assay protocol was used as described by Simon et al. [24], with minor modifications. Briefly, 200 µL of normal melting point agarose gel (0.5%; Tebu-Bio, Heerhugowaard, The Netherlands) was pipetted onto prewarmed glass slides (45 °C) and covered with a 24 mm by 50 mm cover slip. Agarose was solidified by maintaining the slides at room temperature for 20 min, and the cover slip was then gently removed. Ten microliters of sperm suspension (snap-frozen samples thawed and diluted to 6 × 10^6^/mL in PBS) was mixed with 75 µL of low melting point agarose (0.5%; Thermo Fisher Scientific) and pipetted on top of the normal melting point agarose. A cover slip was quickly placed, and the slides were left at room temperature for 20 min. After removing the cover slip, the slides were immersed in 45-mL lysis solution (2.5-M NaCl, 100-mM Na_2_EDTA and 10-mM Tris–HCl, kept at 4 °C and mixed with 500 µL of Triton X-100) and kept for 1 h at 4 °C in a coplin jar. The slides were removed, and 2.5 mL of Dithiothreitol (10mM) was added, and the coplin jar was inverted to ensure mixing. The slides were immersed again and kept for 30 min at 4 °C. After removing the slides, 2.5 mL of Lithium di-iodosalicylate (4 mM) were added into the coplin jar, and, after mixing, the slides were immersed again and maintained for 90 min at room temperature. The slides were removed and drained by standing them vertically on tissue paper and then immersed in fresh alkaline electrophoresis buffer (15 mL of 10-M NaOH and 2.5 mL of 200-mM EDTA in 1000-mL deionized distilled water) for 20 min. Electrophoresis was performed for 10 min by applying a 25-V current adjusted to 300 ± 4 mA. After electrophoresis, the slides were removed, drained as previously and flooded with three changes of neutralization buffer (0.4-M Tris) for 5 min each. The slides were drained again and stained with 50 µL of freshly prepared 1:9 diluted Ethidium bromide staining solution (stock solution 200mg/mL; Thermo Fisher Scientific) and covered with a cover slip. The slides were visualized with a TCS SPE-II system (Leica Microsystems GmbH, Wetzlar, Germany) attached to an inverted semiautomated DMI4000 microscope (Leica) with a 20× magnification objective (oil); images were saved for the subsequent analysis. Fifty comets per slide (Figure 2) were analyzed using the Open comet tool [25]. The following comet parameters were recorded for analysis: (1) tail DNA% (tail DNA content as a percentage of the total comet DNA content), (2) tail moment (tail length multiplied by tail DNA%), (3) olive moment (product of tail DNA% and the distance between the intensity-weighted centroids of the head and tail) and (4) tail length. Spermatozoa treated with DNase II (1mg/mL) served as a positive control and were included in each electrophoresis run.

### 2.5. Modified Flotation DGC Processing of Ejaculates from Friesian Stallions with High Sperm DFI

Semen was collected on alternate days from 3 mature Friesian stallions (10 ± 6 years) with high sperm DFI (range: 22–41%). In total, three ejaculates per stallion were processed using the modified flotation DGC with Opti-prep^TM^ 20% as the top layer (Opti-prep^TM^ 40%/DS-Opti-prep^TM^/Opti-prep^TM^ 20%), and sperm recovery, motility, morphology and DNA integrity (SCSA and comet assay) were determined as described earlier for Band 2 (interface between Opti-prep^TM^ 20% and DS-Opti-prep^TM^). In addition, viability and acrosome integrity were assessed along with mitochondrial and total cellular ROS production using MitoSOX Red (MSR; Thermo Fisher Scientific) and dihydroethidium (DHE; Thermo Fisher Scientific) as described by Hall et al. [26], with some minor modifications. Briefly, 200 μL of sperm suspension in HBSS (2 million spermatozoa) were incubated at 37 °C for 15 min with Hoechst 33258 (H33258: final concentration, 1 µg/µL), PNA-Alexa Flour 647 (final concentration, 1 µg/µL) and 2 μM (final concentration) of MSR or DHE. Sperm cells treated with 25-μM arachidonic acid served as a positive control for high MSR or high DHE levels, because this treatment induces ROS generation via the mitochondrial pathway [27]; the arachidonic acid treatment was included in each flow cytometry session/replicate. Blue (H33258), far-red (PNA-AF647) and red (MSR or DHE) fluorescence were collected using the 450/50 BP, 660/20 BP and 585/42 BP filters, respectively. After defining the single-cell population, live acrosome-intact spermatozoa were gated for low mitochondrial or low total cell reactive oxygen species (ROS). The results are reported as the percentage of viable acrosome-intact sperm cells with low ROS.

## 3. Statistical Analysis

The Wilcoxon matched pairs signed rank test was used to detect the differences in (a) sperm numbers passing through or collecting at the interfaces of the different Opti-prep^TM^ solutions, (b) semen quality parameters before and after modified sedimentation or modified flotation DGC and (c) semen quality parameters between the three best methods of DGC. A linear mixed model was used to examine the effect of the modified flotation DGC on sperm DFI in ejaculates from Friesian stallions (*n* = 3) with high sperm DFI (*n* = 9 ejaculates). Statistical analysis was performed using JMP^®^ software (version 15.2; SAS Institute, Cary, NC, USA), and a *p*-value ≤ 0.05 was considered statistically significant. Data are reported as the mean ± SD (standard deviation).

## 4. Results

### 4.1. Determination of the Optimal Opti-prep^TM^ Solution to Use as Cushion/Bottom Layer

Sperm passage through the Opti-prep^TM^ layer decreased significantly with an increase in the density of the Opti-prep^TM^ solution (Figure 3). Sperm passage was significantly higher for the 20% and 30% Opti-prep^TM^ solutions than for the 40%, 50% and 60% solutions and was significantly higher for the Opti-prep^TM^ 20% than the 30% solution. However, no significant difference in sperm passage was evident between the 40%, 50% and 60% Opti-prep^TM^ solutions. Therefore, an Opti-prep^TM^ 40% solution was used in the subsequent experiments as the cushion/bottom layer.

### 4.2. Modified Sedimentation and Modified Flotation DGC Techniques

#### 4.2.1. Modified Sedimentation DGC

The modified sedimentation density gradient centrifugation technique yielded two bands (sperm layers): one band at the interface between the diluted semen and Opti-prep^TM^ top layer (Band1) and a second band at the interface between the Opti-prep^TM^ top and bottom layers (Band2) (Figure 4a). The sperm recovery and quality parameters for all of the bands are presented in Table 1. The sperm cell recovery was highest in Band2 for Opti-prep^TM^ 20% and 25% and in Band1 for Opti-prep^TM^ 30% (Table 1), and these were therefore considered the most interesting for comparing the effects on sperm cell quality. The percentage of spermatozoa with an intact plasma membrane and acrosome was increased in Band1 for Opti-prep^TM^ 25% and 30%. The percentage of sperm with normal morphology was increased in Band2 compared to non-centrifuged extended semen (NC) for all the top layer Opti-prep^TM^ concentrations tested and did not differ between the NC and Band1 for Opti-prep^TM^ 30%. However, the total motility was increased above the NC only in Opti-prep^TM^ 30% Band1, while the percentage of progressively motile spermatozoa did not increase above the NC in any of the post-DGC bands. The DFI was significantly lower in Band1 for Opti-prep^TM^ 30% than in raw semen but did not differ between raw semen and either Band1 or Band2 for Opti-prep^TM^ 20% and 25%. With regards to the bands of interest (Band2 Opti-prep^TM^ 20% and 25% and Band1 Opti-prep^TM^ 30%), the sperm DFI significantly decreased and percentages of viable acrosome-intact and (total) motile sperm increased in Band1 for Opti-prep^TM^ 30% using the modified sedimentation DGC. This approach was therefore considered the best for modified sedimentation DGC and used for the subsequent comparison with the modified flotation DGC.

#### 4.2.2. Modified Flotation DGC

The modified flotation DGC yielded three bands (selected sperm populations): Band1 (on the Opti-prep^TM^ top layer), Band 2 (at the interface between the Opti-prep^TM^ top layer and the DS-Opti-prep^TM^) and Band3 (interface between DS-Opti-prep^TM^ and the Opti-prep^TM^ bottom layer) (Figure 4b). The sperm recovery was the highest in Band2 for Opti-prep^TM^ 20% and 25% and Band1 for Opti-prep^TM^ 30% (Table 2). On the basis of a higher sperm yield, we focused on these three bands for their sperm quality parameters. The percentage of viable and acrosome-intact spermatozoa was significantly improved above the NC in Band2 for Opti-prep^TM^ 20% and 25%. The total and progressive motility were significantly increased in Band2 for all the Opti-prep^TM^ top layers tested. The DFI was significantly lower than in snap-frozen raw semen in Band1 and Band2 for nearly all the Opti-prep^TM^ top layers examined (except Band2 for Opti-prep^TM^ 30%) and was significantly higher in Band3 for all the top layers. The percentage of sperm cells with normal morphology was significantly higher in Band2 with Opti-prep^TM^ 20% and 25%. Overall, the DFI was reduced in Band2 for Opti-prep^TM^ 20% and 25% (Table 2), and the percentages of viable acrosome-intact, total and progressively motile, and morphology normal sperm all increased in Band2 for Opti-prep^TM^ 20% and 25% compared to the uncentrifuged samples. Therefore, Band2 from Opti-prep^TM^ 20% and 25% were chosen for further comparisons with the best variant from the modified sedimentation DGC.

#### 4.2.3. Comparison between the Modified Sedimentation and Modified Flotation DGC Techniques

A comparison was made between one modified sedimentation DGC (Band1 from Opti-prep^TM^ 30%) and two modified flotation DGC (Band2 from Opti-prep^TM^ 20% and 25%) protocols to determine which method most improved the semen quality parameters (Table 3). One of the modified flotation DGCs (Band2 from Opti-prep^TM^ 20%) and the modified sedimentation DGC (Band1 with Opti-prep^TM^ 30%) protocol resulted in similar sperm recoveries (57 ± 7% and 63 ± 11%, respectively) that were much higher than for the other modified flotation DGC protocol (Band2 from Opti-prep^TM^ 25%: 37 ± 14%; Table 3). All three protocols resulted in significant increases in the percentages of the (total) motile sperm cells and spermatozoa with an intact plasma membrane and acrosome compared to the NC and a significant drop in the DFI compared to raw semen, with no differences between the protocols. By contrast, the percentages of progressively motile spermatozoa and sperm cells with normal morphology were significantly increased above the NC by both modified flotation DGC (Band2 with Opti-prep^TM^ 20% and 25%) but not the modified sedimentation DGC (Band1 with Opti-prep^TM^ 30%) protocol. Overall, although all three protocols significantly improved the DFI in the selected sperm population, the modified flotation DGC protocol using Opti-prep^TM^ 20% as the top layer was preferred on the basis of better sperm recovery than the modified flotation DGC with an Opti-prep^TM^ 25% top layer and higher percentages of progressively motile and morphologically normal sperm than the modified sedimentation DGC protocol with the 30% top layer. The modified flotation DGC with Opti-prep^TM^ 20% was therefore selected to process the ejaculates from three Friesian stallions with high sperm DFI.

#### 4.2.4. Modified Flotation DGC Processing of Ejaculates from Friesian Stallions with High Sperm DFI

The average sperm recovery in Band2 after modified flotation DGC with an Opti-prep^TM^ 20% top layer was only 20 ± 3% (Table 4). The percentages of viable acrosome-intact sperm cells with low mitochondrial and low total cellular ROS were significantly higher after (83 ± 8% and 77 ± 8%, respectively) than the prior-to-modification flotation DGC (62 ± 8% and 60 ± 6%, respectively). In addition, the modified flotation DGC resulted in significant increases compared to pre-centrifugation samples in total motility (TM; 66 ± 10% post- vs. 47 ± 7% pre-DGC, respectively), progressive motility (PM; 50 ± 14% post- vs. 23 ± 10% pre-DGC) and percentage of spermatozoa with normal morphology (NM; 49 ± 8% post- vs. 37 ± 10%). Most importantly, the sperm DFI as measured by SCSA was much lower after (5 ± 2%) than prior to (31 ± 11%) the modified flotation DGC. Similarly, the comet assay parameter, %tail DNA, was significantly reduced after the modified flotation DGC (38 ± 3%) compared to in raw semen (57 ± 3%). Finally, there were no significant differences in the percentage of sperm with low mitochondrial or total ROS levels within the viable, acrosome-intact sperm population.

## 5. Discussion

The aim of the current study was to optimize a DGC technique for removing DNA fragmented spermatozoa from the semen of stallions with high DFI levels, without any adverse effects on the other aspects of sperm quality or excessive loss of “normal” sperm. The modified flotation DGC protocol that involved mixing extended equine semen with Opti-prep^TM^ (Iodixanol 60%) to increase its density prior to layering between the gradient solutions of lower and higher density does not appear to have been reported previously for stallion semen. This modified flotation DGC protocol proved superior to the modified sedimentation DGC in that, while both significantly reduced the DFI of the selected sperm cell population, only the modified flotation DGC reliably increased the percentages of (progressively) motile spermatozoa and spermatozoa with normal morphology. Moreover, the modified flotation DGC appears to be useful for processing ejaculates of (Friesian) stallions with high sperm DFI (>20%), because it reduced the mean sperm DFI from 31% to 5% while simultaneously improving the other semen quality parameters, including the percentages of viable, acrosome-intact, motile, progressively motile and morphologically normal spermatozoa.

Opti-prep^TM^ 40% was selected as the most suitable density to serve as a cushion layer, because only a small percentage (4%) of sperm cells passed through this solution. This suggests that the majority of stallion spermatozoa had a density of less than 1.215 g/mL. The selected cushion layer contrasts to a previous study in which Opti-prep^TM^ 30% was used as a bottom layer [12]. Although the same centrifugation force and time were used in both studies (1000× *g*, 20 min), we compared different density gradient solution percentages (Opti-prep^TM^ 20–60%) to establish the optimal bottom layer. Opti-prep^TM^ 30% appeared to be less suitable as a bottom layer in our study, because approximately 20% of the sperm cells passed through the cushion during centrifugation.

The modified flotation DGC yielded three different sperm populations, compared to the two different sperm populations resulting after the modified sedimentation DGC protocol. Clearly, this is related to the number of colloid density interfaces created and yielding three populations via the modified flotation DGC replicated the results of a study on bull semen [8]. Although the colloid densities used in this study and the exact layering procedure differed between the current study and the bull semen study [8], this was a requirement of the difference in density between stallion and bull spermatozoa. Nevertheless, this suggests that the modified flotation DGC allows a more accurate separation of spermatozoa into different populations with distinct densities. This was apparent both for stallions of various breeds with grossly normal semen parameters and the subfertile Friesian stallions. The modified flotation DGC selected higher quality spermatozoa without any obvious adverse effects on the other parameters of sperm quality. The yield of high-quality spermatozoa was higher than reported for double-layer DGC in previous studies using Opti-prep^TM^ [12], Percoll [14] or EquiPure [28]. However, the improvement in DFI in the recovered spermatozoa in our study was comparable to previous studies [12,28].

The three sperm populations separated by the modified flotation DGC technique were separated based on the spermatozoa’s density. It was obvious that a majority of morphologically defective spermatozoa were less dense and moved upwards to the top of the Opti-prep^TM^ layer with the lowest density (20% or 25%), indicating that they had densities of less than 1.110 g/mL or 1.137 g/mL. The majority of the progressively motile and morphologically normal spermatozoa with intact DNA also moved upwards but stayed at the interface between the low-density Opti-prep^TM^ and the layer of diluted semen mixed with Opti-prep^TM^. The density of the mixed layer was theoretically close to a 30% Opti-prep^TM^ layer. Therefore, the density of the cells must range between 1.110 g/mL and 1.163 g/mL. Nonviable (membrane-damaged) spermatozoa and spermatozoa with a high DFI were generally dense and, therefore, stayed at the interface of semen mixed with Opti-prep^TM^ and the cushion layer after centrifugation [8].

Among the three most promising DGC variants selected for direct comparison, the modified flotation DGC with an upper layer of Opti-prep^TM^ 20% appeared to be the best, because it combined improvement in the range of semen quality parameters with a relatively high sperm recovery. Using Opti-prep^TM^ 25% as the top layer in a modified flotation DGC protocol similarly improved many of the semen quality parameters and seemed to select a population with better progressive motility than Opti-prep^TM^ 20%; however, the sperm recovery was almost half that recorded for Opti-prep^TM^ 20%. This would be a disadvantage if the spermatozoa were being collected for insemination, because it would lead to a smaller dose, or fewer insemination doses, and might therefore negatively influence the number of mares that could be inseminated and become pregnant. On the other hand, the modified flotation DGC with Opti-prep^TM^ 25% could be beneficial when selecting small numbers of high-quality spermatozoa for assisted reproductive technologies, such as intracytoplasmic sperm injection. Although the sperm recovery and DFI appeared to be similar between the modified sedimentation DGC with Opti-prep^TM^ 30% and the modified flotation DGC with Opti-prep^TM^ 20%, the percentages of progressively motile and morphologically normal sperm were lower after the modified sedimentation DGC. This would result in a lower number of high-quality spermatozoa available for insemination.

This study indicated that the modified flotation DGC could have clinical value for processing ejaculates of stallions with high sperm DFI levels (>20–30%). Given that sperm DFI levels > 25% have been reported to compromise fertility [29] and the defect is non-compensable, it is imperative to decrease the percentage of spermatozoa with damaged DNA prior to insemination by applying a sperm selection technique. The modified flotation DGC was effective in markedly reducing the percentage of sperm with fragmentated DNA in ejaculates from Friesian stallions, from levels considered very likely to compromise fertility (30%) to levels compatible with normal-to-good fertility (5%). While the SCSA quantifies the ratio of single-stranded to double-stranded DNA in a single cell, the alkaline comet assay detects the extent of single- and double-stranded breaks, incomplete excision repair sites, crosslinks and alkaline labile sites in the DNA (reviewed by Kumaravel et al. [30]). In analogy to the SCSA, the alkaline comet assay showed that the modified flotation DGC enriched for cells with fewer average DNA damage/labile sites (37% tail DNA) than in the original sample (non-selected sperm: 57% tail DNA). Nevertheless, it remains to be determined to what extent selecting spermatozoa using this modified flotation DGC improves the fertility of these stallions after artificial insemination. While the modified flotation DGC also improved the other semen quality parameters (i.e., viability, acrosome integrity, total motility, progressive motility and normal morphology), the sperm recovery by DGC of semen with a high DFI was low (approximately 20%). This low sperm recovery could, however, be considered a predictable consequence of the high percentage of DNA-damaged and morphologically abnormal spermatozoa present in the ejaculates of stallions with a high sperm DFI. Processing the whole ejaculates for the three Friesian stallions included in this study would have resulted in total mean yields of 222 × 10^6^, 118 × 10^6^ and 382 × 10^6^ morphologically normal, motile sperm per ejaculate. This low sperm recovery may therefore mean that low dose deep-intrauterine insemination close to ovulation needs to be performed to compensate for low insemination doses after applying this modified flotation DGC (a standard dose for insemination with fresh semen is 300 × 10^6^ morphologically normal, motile sperm [31]). Certainly, further studies are required to upscale this modified flotation DGC to process whole ejaculates for AI, since, in the current study, only a portion of the ejaculate was processed.

Finally, the semen samples from the Friesian stallions were also evaluated for mitochondrial and total cellular reactive oxygen species (ROS) prior to and after centrifugation. It has been suggested that the physical shearing forces associated with sperm centrifugation can trigger ROS generation [32]. However, we did not detect a significant increase in viable spermatozoa with elevated ROS levels after centrifugation, indicating that this modified flotation DGC has no indirect detrimental effect on the oxidative status of stallion spermatozoa. The stepwise development of the modified flotation DGC methods presented in this manuscript can likewise be transferred to semen from other species and facilitate improvements in the density gradient-based selection of sperm subpopulations.

## 6. Conclusions

In conclusion, a modified flotation density gradient centrifugation protocol in which semen was mixed 1:1 with Opti-prep^TM^ 60% and layered between an Opti-prep^TM^ 20% top layer and Opti-prep^TM^ 40% bottom layer was an effective method for decreasing the percentage of sperm cells with fragmentated DNA and increasing the percentage that were viable, acrosome-intact, motile and morphologically normal. While the number of recovered spermatozoa from stallions with initially poor semen quality may be limiting for conventional insemination, it should be adequate for deep intrauterine insemination and ample for ICSI.

## Figures and Tables

**Figure 1 animals-11-01973-f001:**
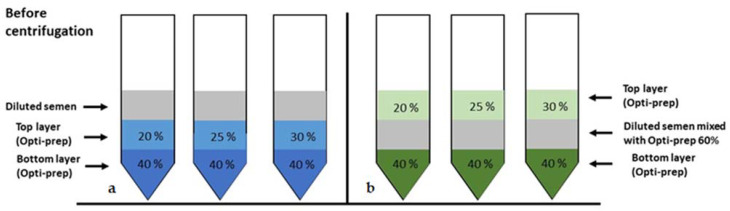
Graphical illustration of the density gradient centrifugation (DGC) techniques before centrifugation at 1000× *g* for 20 min. (**a**) Modified sedimentation DGC technique before centrifugation. (**b**) Modified flotation DGC centrifugation technique before centrifugation.

**Figure 2 animals-11-01973-f002:**
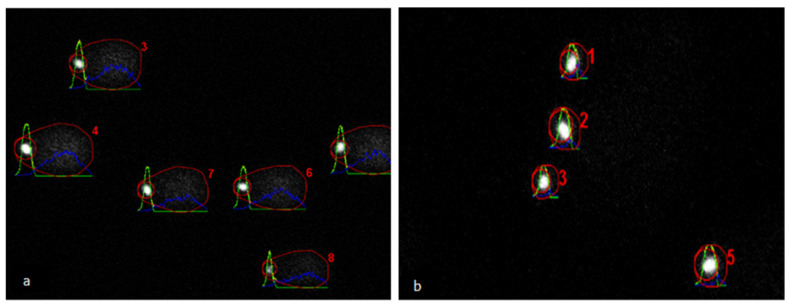
Images of the alkaline comet assay analyzed using Open comet software. (**a**) Raw semen sample containing DNA-damaged spermatozoa. (**b**) Band2 from a modified flotation DGC (top layer Opti-prep^TM^ 20% diluted semen mixed with Opti-prep^TM^ 60% and bottom layer Opti-prep^TM^ 40%) with spermatozoa without DNA damage. Red outline and number (regular comet suitable for evaluation), green profile (comet head), yellow profile (comet tail) and blue profile (comet equal to head plus tail) (full details at https://cometbio.org/documentation.html#interface, accessed on 30 June 2021).

**Figure 3 animals-11-01973-f003:**
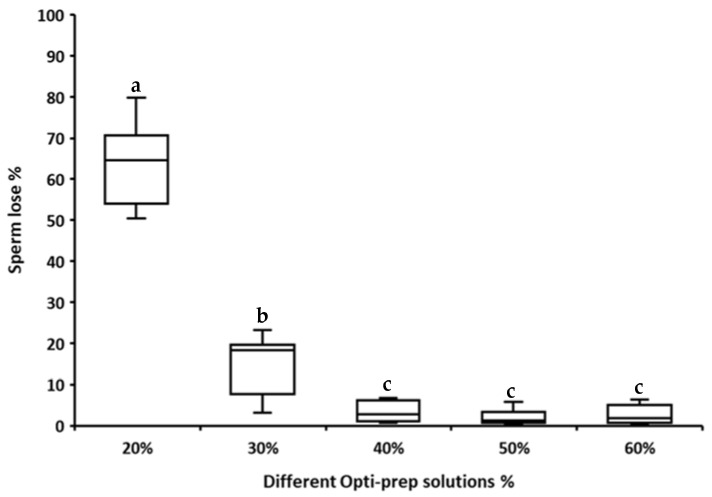
Percentage of sperm lost through the cushion layer after centrifugation at 1000× *g* for 20 min using cushion layers consisting of different percentages of Opti-prep^TM^ (20–60%). The lines (“whiskers”) on the top and bottom of each box show the range of sperm lost, and the horizontal line on each box represents the median (*n* = 6 stallions). Different letters indicate boxes that differ significantly (*p* ≤ 0.05).

**Figure 4 animals-11-01973-f004:**
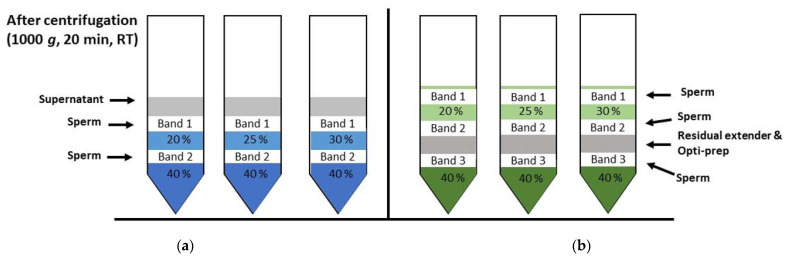
Graphical illustration of the density gradient centrifugation (DGC) techniques after centrifugation at 1000× *g* for 20 min. (**a**) Modified sedimentation DGC technique after centrifugation. (**b**) Modified flotation DGC technique after centrifugation.

**Table 1 animals-11-01973-t001:** Effect of modified sedimentation density gradient centrifugation (DGC) on stallion sperm populations. Semen parameters include sperm recovery (SR), viability and acrosome intactness (VAI), total motility (TM), progressive motility (PM), average path velocity (VAP), curvilinear velocity (VCL), straight line velocity (VSL), amplitude of lateral head displacement (ALH), beat cross-frequency (BCF), wobble (WOB), straightness (STR), linearity (LIN), DNA fragmentation index (DFI) and normal morphology (NM) before and after centrifugation by the modified sedimentation DGC (diluted semen layered onto the Opti-prep^TM^ layers). Within a row, values with different superscript letters differ significantly (*p* ≤ 0.05, *n* = 7 stallions).

Semen Parameter	Non-Centrifuged Semen	Opti-prep^TM^ 20% Top Layer		Opti-prep^TM^ 25% Top Layer		Opti-prep^TM^ 30% Top Layer	
		Band1	Band2	Band1	Band2	Band1	Band2
SR %		9 ± 4 ^a^	66 ± 14 ^b^	32 ± 16 ^c^	46 ± 13 ^c,d^	63 ± 11 ^b,d^	16 ± 6 ^a^
VAI %	81 ± 6 ^a^	87 ± 8 ^a,c^	62 ± 19 ^b^	89 ± 6 ^c^	52 ± 19 ^b^	90 ± 6 ^c^	20 ± 17 ^d^
TM %	69 ± 7 ^a^	43 ± 10 ^b,c^	64 ± 12 ^a^	65 ± 10 ^a,c^	55 ± 16 ^c^	80 ± 8 ^d^	31 ± 21 ^b^
PM %	50 ± 16 ^a^	25 ± 8 ^b,c^	26 ± 9 ^b^	47 ± 4 ^a^	18 ± 9 ^c^	55 ± 11 ^a^	9 ± 12 ^d^
VAP μm/s	99 ± 10 ^a^	89 ± 9 ^a,c^	64 ± 13 ^b,d^	84 ± 20 ^c,d^	60 ± 16 ^b,d^	87 ± 15 ^c,d^	65 ± 20 ^d^
VCL μm/s	157 ± 19 ^a^	157 ± 14 ^a^	105 ± 23 ^b,c^	150 ± 39 ^a,c^	103 ± 27 ^c^	140 ± 19 ^c^	113 ± 32 ^c^
VSL μm/s	78 ± 11 ^a^	67 ± 8 ^a,c^	52 ± 12 ^b,c,e^	65 ± 14 ^c,e^	49 ± 15 ^d,e^	70 ± 12 ^c,e^	51 ± 18 ^e^
ALH μm	3.3 ± 0.6 ^a^	3 ± 0.4 ^a,c^	2.4 ± 0.5 ^b,c^	3 ± 0.8 ^a,b,c^	2.3 ± 0.4 ^b,c^	2.9 ± 0.5 ^a,b,c^	2.5 ± 0.8 ^c^
BCF Hz	42 ± 4 ^a^	41 ± 3 ^a^	38 ± 5 ^a^	40 ± 3 ^a^	37 ± 5 ^a^	41 ± 3 ^a^	38 ± 6 ^a^
WOB %	63 ± 5 ^a,c^	56 ± 7 ^b,c^	61 ± 9 ^a,b,c^	57 ± 7 ^b^	58 ± 6 ^a,b,c^	62 ± 5 ^c^	57 ± 10 ^a,b,c^
STR %	80 ± 7 ^a^	80 ± 6 ^a^	80 ± 8 ^a^	80 ± 6 ^a^	80 ± 7 ^a^	80 ± 6 ^a^	80 ± 11 ^a^
LIN %	50 ± 8 ^a,b^	43 ± 8 ^a,b^	50 ± 12 ^a,b^	44 ± 8 ^b^	47 ± 9 ^a,b^	50 ± 7 ^a,b^	45 ± 14 ^a,b^
DFI %	12 ± 6 ^a^	10 ± 11 ^a^	10 ± 3 ^a^	10 ± 9 ^a,b^	14 ± 7 ^a^	5 ± 3 ^b^	33 ± 17 ^c^
NM %	52 ± 12 ^a^	16 ± 12 ^b^	62 ± 18 ^c^	24 ± 10 ^b^	74 ± 10 ^d^	47 ± 15 ^a^	67 ± 14 ^e^

**Table 2 animals-11-01973-t002:** Effect of a modified flotation density gradient centrifugation (DGC) protocol on stallion sperm populations. The semen parameters include sperm recovery (SR), viability and acrosome intactness (VAI), total motility (TM), progressive motility (PM), average path velocity (VAP), curvilinear velocity (VCL), straight line velocity (VSL), amplitude of lateral head displacement (ALH), beat cross-frequency (BCF), wobble (WOB), straightness (STR), linearity (LIN), DNA fragmentation index (DFI) and normal morphology (NM) before and after centrifugation by the modified flotation DGC (diluted semen mixed with Opti-prep^TM^ 60% and layered between the top layer and a 40% Opti-prep^TM^ bottom layer). Within a row, values with different superscript letters differ significantly (*p* ≤ 0.05, *n* = 7 stallions).

Semen Parameter	Non-Centrifuged Semen	Opti-prep^TM^ 20% Top Layer			Opti-prep^TM^ 25% Top Layer			Opti-prep^TM^ 30% Top Layer		
		Band1	Band2	Band3	Band1	Band2	Band3	Band1	Band2	Band3
SR %		8 ± 6 ^a,e^	57 ± 7 ^b,d^	10 ± 2 ^a,e^	28 ± 14 ^c^	37 ± 14 ^c,d^	10 ± 2 ^a^	47 ± 16 ^d^	21 ± 10 ^e^	9 ± 1 ^a^
VAI %	81 ± 4 ^a^	52 ± 31 ^b,d^	89 ± 4 ^c^	27 ± 7 ^b^	67 ± 16 ^d^	88 ± 5 ^c^	25 ± 9 ^b^	85 ± 4 ^a,c^	85 ± 6 ^a,c^	25 ± 10 ^b^
TM %	69 ± 7 ^a^	33 ± 14 ^b^	82 ± 10 ^c,e^	30 ± 13 ^b,d,f^	52 ± 13 ^d^	83 ± 10 ^e^	30 ± 13 ^b,d,f^	66 ± 10 ^f^	77 ± 16 ^c,e^	27 ± 14 ^d^
PM %	50 ± 16 ^a^	9 ± 6 ^b^	62 ± 10 ^c,g^	21 ± 13 ^b,d,f^	19 ± 11 ^d^	72 ± 9 ^e,g^	22 ± 12 ^b,d,f^	31 ± 9 ^f^	70 ± 16 ^g^	19 ± 13 ^b,d^
VAP μm/s	99 ± 10 ^a^	71 ± 21 ^b,d,e^	90 ± 7 ^c,d^	73 ± 11 ^b,d,e^	79 ± 19 ^d,e^	88 ± 17 ^a,d^	78 ± 19 ^a,d,e^	69 ± 11 ^e^	91 ± 7 ^c,d^	64 ± 15 ^e^
VCL μm/s	157 ± 19 ^a^	137 ± 41 ^a,b,c^	142 ± 19 ^a^	115 ± 31 ^b,c^	142 ± 27 ^a^	134 ± 33 ^a,b^	117 ± 37 ^a,b,c^	122 ± 20 ^b,c^	139 ± 23 ^a^	97 ± 33 ^c^
VSL μm/s	78 ± 11 ^a^	53 ± 14 ^b^	72 ± 6 ^c^	60 ± 5 ^b^	57 ± 11 ^b,c^	74 ± 13 ^a,c^	65 ± 12 ^a,b,c^	53 ± 9 ^b^	76 ± 6 ^a^	55 ± 9 ^b^
ALH μm	3.3 ± 0.6 ^a,d^	3.1 ± 0.8 ^a,d^	2.8 ± 0.6 ^a^	2.5 ± 0.9 ^a,d^	3.2 ± 0.7 ^d,e^	2.5 ± 0.7 ^e^	2.2 ± 0.8 ^b,e^	2.7 ± 0.5 ^c^	2.6 ± 0.7 ^c^	2 ± 0.8 ^c^
BCF Hz	42 ± 4 ^a,b,c,d^	37 ± 3 ^a,d^	42 ± 2 ^b,d^	39 ± 3 ^a,d^	39 ± 2 ^a^	43 ± 2 ^c,d^	42 ± 3 ^d^	38 ± 2 ^a^	42 ± 3 ^b^	41 ± 2 ^b^
WOB %	63 ± 5 ^a,c,d^	51 ± 6 ^b,c^	63 ± 7 ^a,d^	64 ± 8 ^a,b,c,d^	54 ± 5 ^b,c,d^	66 ± 6 ^a,d^	67 ± 6 ^a,d^	56 ± 5 ^c^	66 ± 8 ^d^	67 ± 8 ^a,d^
STR %	80 ± 7 ^a,e^	80 ± 9 ^a,b,c,e^	80 ± 8 ^a,c,d,e^	80 ± 8 ^a,c,d,e^	70 ± 5 ^b^	80 ± 7 ^c^	90 ± 7 ^e^	80 ± 7 ^a^	80 ± 8 ^e^	90 ± 6 ^d,e^
LIN %	50 ± 8 ^a,b,d^	40 ± 8 ^a,c,d^	50 ± 1 ^b^	60 ± 1 ^a,b,c^	40 ± 3 ^c,d^	60 ± 9 ^b^	60 ± 9 ^b^	40 ± 7 ^d^	60 ± 1 ^b^	60 ± 1 ^b^
DFI %	12 ± 6 ^a,e^	3 ± 2 ^b,d^	4 ± 2 ^b^	41 ± 13 ^c^	3 ± 2 ^b,d^	5 ± 4 ^b,e^	42 ± 17 ^c^	3 ± 1 ^d^	7 ± 3 ^e^	40 ± 18 ^c^
NM %	52 ± 12 ^a^	10 ± 5 ^b^	66 ± 16 ^c,d^	65 ± 9 ^a,c,d^	13 ± 7 ^b^	73 ± 7 ^d,f^	63 ± 10 ^c^	33 ± 15 ^e^	76 ± 9 ^f^	58 ± 14 ^a,c,d^

**Table 3 animals-11-01973-t003:** Comparing the effects of different density gradient centrifugation (DGC) protocols on stallion sperm populations. The semen parameters include sperm recovery (SR), viability and acrosome intactness (VAI), total motility (TM), progressive motility (PM), DNA fragmentation index (DFI) and normal morphology (NM) before and after centrifugation by the modified sedimentation DGC (Band1 with 30% Opti-prep^TM^) and modified flotation methods of DGC (Band2 with 20% and 25% Opti-prep^TM^). Different letters indicate significant differences (*p* ≤ 0.05, *n* = 7 stallions).

Semen Parameter	Diluted Semen (or Raw: DFI)	20% Opti-prep^TM^ (Modified Flotation DGC)	25% Opti-prep^TM^ (Modified Flotation DGC)	30% Opti-prep^TM^ (Modified Sedimentation DGC)
		Band2	Band2	Band1
SR %		57 ± 7 ^a^	37 ± 14 ^b^	63 ± 11 ^a^
VAI %	81 ± 4 ^a^	89 ± 4 ^b^	88 ± 5 ^b^	90 ± 6 ^b^
TM %	69 ± 7 ^a^	82 ± 10 ^b^	83 ± 10 ^b^	80 ± 8 ^b^
PM %	50 ± 16 ^a^	62 ± 10 ^b^	72 ± 9 ^c^	55 ± 11 ^a^
DFI %	12 ± 6 ^a^	4 ± 2 ^b^	5 ± 4 ^b^	5 ± 3 ^b^
NM %	52 ± 12 ^a^	66 ± 16 ^b^	73 ± 7 ^b^	47 ± 15 ^a^

**Table 4 animals-11-01973-t004:** Effect of a modified flotation density gradient centrifugation (DGC) protocol on the sperm recovery (SR), viable acrosome-intact spermatozoa with low mitochondrial reactive oxygen species (VAI L-M-ROS), viable acrosome-intact spermatozoa with low total cellular reactive oxygen species (VAI L-T-ROS), total motility (TM), progressive motility (PM), DNA fragmentation index (DFI), comet assay tail length, tail DNA, tail moment, olive moment and normal morphology (NM) for 3 Friesian stallions (3 ejaculates per stallion, *n* = 9) with a high initial DFI. Within a row, an asterisk indicates values that differ before and after the modified flotation DGC (*p* ≤ 0.05).

Semen Parameter	Stallion 1		Stallion 2		Stallion 3		Overall Average	
	Before	After	Before	After	Before	After	Before	After
SR%		18 ± 10		25 ± 6		20 ± 3		
VAI L-M-ROS%	56 ± 8	75 ± 10	63 ± 9	87 ± 5	66 ± 1	86 ± 3	62 ± 8	83 ± 8 *
VAI L-T-ROS %	54 ± 5	74 ± 6	59 ± 2	78 ± 3	66 ± 4	80 ± 13	60 ± 6	77 ± 8 *
TM%	42 ± 3	56 ± 8	45 ± 9	61 ± 1	54 ± 3	75 ± 3	47 ± 7	66 ± 10 *
PM%	12 ± 4	38 ± 6	24 ± 5	43 ± 8	34 ± 2	64 ± 5	23 ± 10	50 ± 14 *
DFI%	41 ± 15	7 ± 3	22 ± 2	4 ± 2	30 ± 2	4 ± 1	31 ± 11	5 ± 2 *
Tail length µm	93 ± 5	43 ± 16	33 ± 31	15 ± 19	74 ± 10	70 ± 6	67 ± 13	42 ± 26 *
Tail DNA%	65 ± 3	32 ± 11	26 ± 10	13 ± 4	80 ± 4	69 ± 9	57 ± 25	38 ± 26 *
Tail moment µm	63 ± 5	17 ± 7	4 ± 3	4 ± 3	59 ± 11	51 ± 3	42 ± 29	24 ± 21
Olive moment µm	39 ± 4	9 ± 4	2 ± 2	3 ± 2	37 ± 8	32 ± 2	26 ± 18	15 ± 13
NM%	25 ± 6	39 ± 3	37 ± 2	52 ± 2	46 ± 2	55 ± 6	37 ± 10	49 ± 8 *

## Data Availability

The data presented in this study are available in A Modified Flotation Density Gradient Centrifugation Technique Improves the Semen Quality of Stallions with a High DNA Fragmentation Index, Umair et al., 2021.

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
