# Peer review of "A Modified Flotation Density Gradient Centrifugation Technique Improves the Semen Quality of Stallions with a High DNA Fragmentation Index"

_animals, 2021, doi:10.3390/ani11071973_

Round 1
Reviewer 1 Report
A well written paper with interesting and useful information. I have no further comments.
Author Response
Dear Reviewer,
On behalf of all authors, I thank you for critically reading and reviewing the article.
Kind regards
Muhammad Umair
Corresponding author
Reviewer 2 Report
Dear Authors,
The aim of this study was to develop a density gradient of centrifugation technique using Optiprep™ based on Iodixanol a non-ionic gradient material with relatively low density.
Colloidal centrifugation systems have shown to provide efficient results in increasing stallion sperm quality, removing debris, pathogens and damage or death spermatozoa from ejaculates.
A mayor concern regarding this manuscript is that in the experimental designed the n used was very low (7 ejaculates, one from each stallion ).
This manuscript is reasonable well written. Results are consistent and conclusion are relevant and may contribute to improve the stallion sperm selection.
Below I summarized some comments to address before this can be considered for publication:
Lines 65, 69,70.- Throughout the text it should be indicated if the product used in different density gradient centrifugation systems are Copyright® or trademark ™.
Line 120. - It is not clear what volume of bottom or top layer was used. Is the proportion 1:1:1 or 1:1:2 as it is indicated in line 145?
Line 113. - 1000g is a high centrifugal force, why this force was selected should be indicated and referenced.
Lines 123-127.” Semen was divided into two portions”. How many mL of semen were used in both techniques?
Line 128. - Why semen was diluted with Optiprep 60%?. Other concentrations efficiency has been previously evaluated? Please, explain it, references should be added.
Lines 187 to 201. - This paragraph should be conveniently enumerated
Line 28? Pg12- Modified processing of ejaculates from Friesian stallions with high sperm DFI.- From my point of view , in table 4, data relating to overall average could be removed. When evaluating each horse individually we found an individual response to the different treatments which reflects the characteristic individual sperm response among horse population.
Table 4 should show results of different seminal parameters from horses with high or low DFI (It could use data from horses of previous experiment). Statistical differences should be shown. Results could be seen more clear adding some graphic representation in which every important seminal parameter was compared between stallions with high or low DFI.
Yours faithfully.
Author Response
General comments to the editor and reviewer 2:
The authors thank the section editor and reviewer 2 for critically reading this manuscript and providing constructive criticism. Our response (italicized) to the comments or questions of each individual reviewer can be found below. Changes made in the manuscript are in red font.
Point 1: Lines 65, 69,70.- Throughout the text it should be indicated if the product used in different density gradient centrifugation systems are Copyright® or trademark ™.
Response 1: Opti-prep is trademark ™ and has been modified throughout the text.
Point 2: Line 120. - It is not clear what volume of bottom or top layer was used
Response 2: 2ml of bottom layer and 2 ml of top layer was used (line 154)
Point 3: Is the proportion 1:1:1 or 1:1:2 as it is indicated in line 145?
Response 3: It is a 1:1 dilution (semen diluted with INRA 96 and Opti-prepTM). Changes have been made to manuscript (line 166)
Point 4: Line 113. - 1000g is a high centrifugal force, why this force was selected should be indicated and referenced.
Response 4: It has been shown that high G-forces can be used with a suitable cushion layer. I also added a reference to the manuscript (Line 132)
Point 5: Lines 123-127.” Semen was divided into two portions”. How many mL of semen were used in both techniques?
Response 5: 2.5 ml of raw semen was used in each technique.
Point 6: Line 128. - Why semen was diluted with Opti-prepTM 60%?. Other concentrations efficiency has been previously evaluated? Please, explain it, references should be added.
Response 6: Opti-prepTM 40% was the optimal bottom later. By diluting extended semen with Opti-prepTM 60%, the solution will layer on top of the 40% and below the 20% so that semen could separate in three different populations (bands).
Point 7: Lines 187 to 201. - This paragraph should be conveniently enumerated
Response 7: Heading of the paragraph was combined with the previous section and is corrected.
2.4.3.3. Viability and acrosome integrity (VAI %) (line 245)
Point 8: Line 28? Pg12- Modified processing of ejaculates from Friesian stallions with high sperm DFI.- From my point of view , in table 4, data relating to overall average could be removed. When evaluating each horse individually we found an individual response to the different treatments which reflects the characteristic individual sperm response among horse population.
Table 4 should show results of different seminal parameters from horses with high or low DFI (It could use data from horses of previous experiment). Statistical differences should be shown. Results could be seen more clear adding some graphic representation in which every important seminal parameter was compared between stallions with high or low DFI.
Response 8: with all due respect, the authors believe that table 4 should not include stallions with low DFI. The aim of this experiment was to determine whether this method (modified flotation density gradient centrifugation) could be used to decrease DFI in stallions with high DFI and likely stallions with fertility problems. All the results from stallions with low DFI are already presented in table. Including this data with table 4 will partially be redundant.
Reviewer 3 Report
General
In subsequent hierarchical experiments, the authors develop successfully a modified technique for sedimentation density gradient centrifugation (DGC) , and a modified technique for flotation density gradient centrifugation of stallion sperm.
The study is well written and contains informations that would be of interest for equine AI-practice. The objectives are clear. Results are meaningful (page 15 L 7-15). Number of repetitions (stallions, ejaculate) is relatively low in each experiment, but still sufficiently high to achieve the research goals.
comments:
Protocol for purification and processing of stallion spermatozoa
L 133 2.4.2. Density gradient centrifugation (standard versus modified DGC):
For a better understanding this reviewer recommends to consider the principles of DGC of spermatozoa. Please use correct, established, and common terminology to describe DGC techniques.
Please avoid the term standard technique the correct terminology here should be modified sedimentation DGC. The term standard technique is misleading because in this study a new combination of centrifugation techniques that has not yet been described is shown. In this study the so-called cushioned-centrifugation technique using a 40% iodixanol cushion layer was combined with a double-layer (diluted sperm sample as top layer and either 20%, 25%, and 30% iodixanol bottom layer) sedimentation DGC.
Please avoid the term modified technique please change into modified flotation DGC throughout the manuscript including figures. The term modified technique is misleading because in this study a new combination of centrifugation techniques that has not yet been described is shown. Here the so-called cushioned-centrifugation technique using a 40% iodixanol cushion layer was combined with flotation DGC (1:1:2 mixture of stallion semen, INRA96 and Opti-prep 60% as bottom resp. intermediate layer and either 20%, 25%, and 30% iodixanol as top layer).
Please reword description for “making density gradients” and “density gradient harvesting”, accordingly.
In order to improve the understanding of the methodology for the reader, this reviewer also recommends that Figure 1, submitted as a supplementary file, be incorporated into the manuscript. Ideally, the upper part of the figure (before centrifugation) should appear in the material method section. The lower part (after centrifugation) in the result chapter.
Page 1 L 13-16, Page 4 145-147, and page 15 L 4-10
The essential criticism of this reviewer, if at all, is that in this study control groups - where the new techniques developed in this study are compared with established techniques – are not listed.
This applies in particular to the reference study 8 (Beer-Ljubic et al.), which was used by the authors of the present study as a template. According to Beer-Ljubic et al. high motility (>90%) sperms were isolated in the fraction between the layers of 1.119 (~21% iod.) and 1.154 g/mL (~28% iod.) iodixanol density solutions, and very low motility sperms in the sperm/iodixanol bottom layer (1.1790 g/ml , ~31%iodixanol). The third fraction with deformed sperm were accumulated on top of the top layer.
In the present study the 1:1:2 mixture of stallion semen, INRA96 and Opti-prep 60% serving as bottom resp. intermediate layer was used to separate sperm and harvesting bands of varying sperm quality.
Surely there are great differences between bull and stallion spermatozoa. Nevertheless, comparative analysis of stallion sperm prepared by flotation DGC as described by Beer and Lubic et al. is missing. Maybe there were preliminary experiments done whose result could be listed.
Page 5 L185
Please add a paragraph before 2.4.3.3
page 6 L 258-264 & page 16 L 83-86
L 83 Processing the whole ejaculates ?
Was this done by using 15ml tubes and 2ml DS-Opti-prep layer multiple times?
Page 15 L 20, 34
Typo? Reference 12 would probably have to be listed here instead of 11.
page 15 L 22-24
Provided that the layer designations in the Supplementary Fig1 are correct (?). Optiprep30% was not tested as Bottom Layer in the submitted study. Please recheck.
Author Response
General comments to the editor and reviewer 3:
The authors thank the section editor and all the reviewer for critically reading this manuscript and providing constructive criticism. Our response (italicized) to the comments or questions of each individual reviewer can be found below. Changes made in the manuscript are in red font.
Point 1:L 133 2.4.2. Density gradient centrifugation (standard versus modified DGC):
For a better understanding this reviewer recommends to consider the principles of DGC of spermatozoa. Please use correct, established, and common terminology to describe DGC techniques.
Please avoid the term standard technique the correct terminology here should be modified sedimentation DGC. The term standard technique is misleading because in this study a new combination of centrifugation techniques that has not yet been described is shown. In this study the so-called cushioned-centrifugation technique using a 40% iodixanol cushion layer was combined with a double-layer (diluted sperm sample as top layer and either 20%, 25%, and 30% iodixanol bottom layer) sedimentation DGC.
Please avoid the term modified technique please change into modified flotation DGC throughout the manuscript including figures. The term modified technique is misleading because in this study a new combination of centrifugation techniques that has not yet been described is shown. Here the so-called cushioned-centrifugation technique using a 40% iodixanol cushion layer was combined with flotation DGC (1:1:2 mixture of stallion semen, INRA96 and Opti-prep 60% as bottom resp. intermediate layer and either 20%, 25%, and 30% iodixanol as top layer).
Response 1: The authors agree with the reviewer. Changes have been made throughout the manuscript
Point 2: Please reword description for “making density gradients” and “density gradient harvesting”, accordingly.
Response 2: The authors do not understand what the reviewer is trying to ask? Could the reviewer explain it a bit more in detail?
Point 3: In order to improve the understanding of the methodology for the reader, this reviewer also recommends that Figure 1, submitted as a supplementary file, be incorporated into the manuscript. Ideally, the upper part of the figure (before centrifugation) should appear in the material method section. The lower part (after centrifugation) in the result chapter.
Response 3: Figure has been added to the manuscript (after line 179,395)
Point 4: Page 1 L 13-16, Page 4 145-147, and page 15 L 4-10
The essential criticism of this reviewer, if at all, is that in this study control groups - where the new techniques developed in this study are compared with established techniques – are not listed.
This applies in particular to the reference study 8 (Beer-Ljubic et al.), which was used by the authors of the present study as a template. According to Beer-Ljubic et al. high motility (>90%) sperms were isolated in the fraction between the layers of 1.119 (~21% iod.) and 1.154 g/mL (~28% iod.) iodixanol density solutions, and very low motility sperms in the sperm/iodixanol bottom layer (1.1790 g/ml , ~31%iodixanol). The third fraction with deformed sperm were accumulated on top of the top layer.
In the present study the 1:1:2 mixture of stallion semen, INRA96 and Opti-prep 60% serving as bottom resp. intermediate layer was used to separate sperm and harvesting bands of varying sperm quality.
Surely there are great differences between bull and stallion spermatozoa. Nevertheless, comparative analysis of stallion sperm prepared by flotation DGC as described by Beer and Lubic et al. is missing. Maybe there were preliminary experiments done whose result could be listed.
Response 4: Indeed, the density of stallion semen is different than the density of bull semen (Beer-Ljubic et al.). Therefore, using the same protocol that has been described for bulls did not give similar results . In preliminary experiment, Opti-prep 40% moved to the bottom of the tube after it was layered on top of diluted semen mixed with Opti-prepTM (1:1).
Point 5: Page 5 L185
Please add a paragraph before 2.4.3.3
Response 5: Heading was missing. It has been placed.
2.4.3.3. Viability and acrosome integrity (VAI %) (line 245)
Point 6: page 6 L 258-264 & page 16 L 83-86
L 83 Processing the whole ejaculates ?
Was this done by using 15ml tubes and 2ml DS-Opti-prep layer multiple times?
Response 6: Yes, it was done using 15ml tubes and DS-Opti-prep layer one time for each treatment. Only a well-known part of the ejaculate was processed.
Point 7: Page 15 L 20, 34
Typo? Reference 12 would probably have to be listed here instead of 11.
Response 7: Correction is done (line 21,38 Page 15)
Point 8: page 15 L 22-24
Provided that the layer designations in the Supplementary Fig1 are correct (?). Optiprep30% was not tested as Bottom Layer in the submitted study. Please recheck.
Response 8: Opti-prep 30% was tested as a bottom layer and is presented in Figure 3 and in section 4.1 L 334. It does not appear to be a suitable bottom layer (see figure 3)